# Therapeutic Potential of Exosomes Derived from Diabetic Adipose Stem Cells in Cutaneous Wound Healing of *db/db* Mice

**DOI:** 10.3390/pharmaceutics14061206

**Published:** 2022-06-06

**Authors:** Hsiang-Hao Hsu, Aline Yen Ling Wang, Charles Yuen Yung Loh, Ashwin Alke Pai, Huang-Kai Kao

**Affiliations:** 1Kidney Research Center, Department of Nephrology, Chang Gung Memorial Hospital & Chang Gung University College of Medicine, Taoyuan 333, Taiwan; hsianghao@gmail.com; 2Center for Vascularized Composite Allotransplantation, Chang Gung Memorial Hospital, Taoyuan 333, Taiwan; aline2355@yahoo.com.tw; 3Department of Plastic Surgery, Addenbrooke’s Hospital, Hills Road, Cambridge CB2 0QQ, UK; charles.loh2@meht.nhs.uk; 4Department of Plastic and Reconstructive Surgery, Chang Gung Memorial Hospital & Chang Gung University College of Medicine, Taoyuan 333, Taiwan; drashwinpai@gmail.com

**Keywords:** exosomes, adipose stem cell, cutaneous wound healing

## Abstract

(1) Background: Diabetes impairs angiogenesis and wound healing. Paracrine secretion from adipose stem cells (ASCs) contains membrane-bound nano-vesicles called exosomes (ASC-Exo) but the functional role and therapeutic potential of diabetic ASC-Exo in wound healing are unknown. This study aims to investigate the in vivo mechanistic basis by which diabetic ASC-Exo enhance cutaneous wound healing in a diabetic mouse model. (2) Methods: Topically applied exosomes could efficiently target and preferentially accumulate in wound tissue, and the cellular origin, ASC or dermal fibroblast (DFb), has no influence on the biodistribution pattern of exosomes. In vivo, full-thickness wounds in diabetic mice were treated either with ASC-Exo, DFb-Exo, or phosphate-buffered saline (PBS) topically. ASC-Exo stimulated wound healing by dermal cell proliferation, keratinocyte proliferation, and angiogenesis compared with DFb-Exo and PBS-treated wounds. (3) Results: Diabetic ASC-Exo stimulated resident monocytes/macrophages to secrete more TGF-β1 and activate the TGF-β/Smad3 signaling pathway. Fibroblasts activated by TGF-β1containing exosomes from ASCs initiate the production of TGF-β1 protein in an autocrine fashion, which leads to more proliferation and activation of fibroblasts. TGF-β1 is centrally involved in diabetic ASC-Exo mediated cellular crosstalk as an important early response to initiating wound regeneration. (4) Conclusions: The application of diabetic ASC-Exo informs the potential utility of a cell-free therapy in diabetic wound healing.

## 1. Introduction

Impaired diabetic wound healing is a major morbidity associated with diabetes, which leads to pain, suffering, poor quality of life, and increases in medical costs for patients. Current trends in diabetic wound healing are geared towards cell therapy of various types to revitalize poor blood supplies in affected areas. Since then, mesenchymal stem cell (MSC)-based therapy has also emerged as a promising strategy for treating chronic wounds via tissue repair and immunomodulation [1,2]. Interestingly, paracrine pathways have been mostly recognized to deliver major healing effects through a plethora of chemokines, cytokines, and growth factors generated from the transplanted cells [3,4]. This, coupled with issues related to cellular antigenicity, mutagenicity, oncogenicity, and poor engraftment, has prompted us to explore cell-free therapy for wound healing [5,6,7].

Exosomes, a class of membraned vesicles secreted by cells in response to certain stimuli, contain proteins, nucleic acids, mRNAs, and miRNAs [8,9]. Emerging evidence shows exosomes may serve as paracrine or autocrine effectors by delivering various bioactive molecules, which mediate intercellular communication and may target or modify specific cell types and their functions [10,11,12]. The clear advantages of an exosome-based therapy over cell-based therapies would include trans-host applicability and easy logistical needs in production and storage. Although studies have shown that ASC-Exo could enhance cutaneous wound healing through different signal pathways and mechanisms, ASC-Exo lacks enough evidence to compare its effectiveness between autologous or allogenic origin.

It has been shown that the administration of MSC-derived exosomes has yielded beneficial effects in a variety of animal models of diseases [13,14,15]. While terminally differentiated fibroblasts or fibroblast-keratinocyte mixtures are known to accelerate diabetic wound healing, it is no surprise that the self-renewable ASCs possess the capacity for multi-lineage differentiation with a remarkable ability to enhance cutaneous wound healing [16,17,18,19,20]. Compared to ASCs, ASC-Exo might avoid immunogenicity-associated problems. Till now, ASC-Exo lacked enough evidence to compare its effectiveness between autologous or allogenic origin. However, Lu et al. demonstrated that allogeneic induced pluripotent stem cell-derived exosome provided less effectiveness in wound healing as compared to the autologous origin [21].

The choice of using ASC and ASC-Exo from diabetic mice is to reflect the possible clinical nature of this implication where autologous cell therapy could potentially be used in this subgroup of diabetic patients. Furthermore, others in the literature have also utilized ASCs from the diabetic population and provided its therapeutic potential in insulin resistance or wound healing [22,23]. Nonetheless, the therapeutic potential of diabetic ASC-Exo in cutaneous wounds is still unclear. In our current study, we hypothesized whether exosomes derived from ASCs or dermal fibroblasts of diabetic mice would exhibit differential characteristics and therapeutic efficacy in diabetic wound regeneration.

We isolated exosomes from culture supernatants of adipose stem cells and dermal fibroblasts (DFb) of diabetic mice and verified their content by electron microscope and nanoparticle tracking analysis. We then investigated their functional relevance with respect to the production of chemokines, growth factors, and extracellular matrix (ECM) components in wounds topically treated with diabetic ASC-Exo or DFb-Exo. We specifically focused on the mechanism of TGF-β1 involvement, a multifunctional cytokine important in various signaling pathways, in diabetic ASC-Exo mediated wound healing.

## 2. Materials and Methods

### 2.1. Mice

Homozygous genetically diabetic 10 to 12-week-old, B6.Lepr*^db/db^* mice were obtained from National Laboratory Animal Center (Taipei, Taiwan). All animals were used under approved animal protocols and housed in the Chang Gung Memorial Hospital at Linkou.

### 2.2. Isolation and Identification of Exosomes

ASCs and DFbs of *db/db* mice were isolated as previously described. ASCs or DFbs were cultured in exosome-free Dulbecco’s modified Eagle’s medium (DMEM; Gibco Invitrogen/Life Technologies, Carlsbad, CA, USA) supplemented with 10% fetal bovine serum (FBS; Gibco Invitrogen/Life Technologies, Carlsbad, CA, USA) at 37 °C in a 5% CO_2_ incubator. Culture supernatants were collected from conditioned mediums, including 5% exosome depleted FBS (Thermo Fisher Scientific, Waltham, MA, USA) after 3 days. The supernatants were first centrifuged at 300× *g* for 10 min to remove all cells and any cell debris. The exosome fraction was precipitated from the supernatant with reagents from the Exo-spin exosome purification kit (Cell Guidance Systems, St. Louis, MO, USA). Purification of the exosomes was then performed with centrifuges with the microcentrifuge collection tube where the final elute of purified exosomes was obtained. The morphological analysis of collected exosomes were observed by transmission electron microscopy (Hitachi-7000FA, Tokyo, Japan). The protein expression of exosome-associated general markers including CD9, CD63, and CD81 (Abcam) were analyzed by Western blotting and flow cytometry. The particle size and distribution of *db/db* mice ASC-Exo were identified by NanoSight LM10 (NanoSight, Amesbury, UK).

### 2.3. In Vivo Imaging

First applications of exosomes, as well as a supernatant control, were labeled with octadecyl rhodamine B (R18; PromoCell, Heidelberg, Germany) according to the manufacturer’s protocol. Labeled exosomes and PBS were topically administered to a 1 × 1 cm wound in the dorsum of *db/db* mice. Respective interval bioluminescent images were captured of wounds in anesthetized mice using the In Vivo Imaging System (IVIS) Spectrum (Perkin Elmer, Santa Clara, CA, USA) and analyzed using IVIS imaging software (Perkin Elmer). The fluorescence images for exosome uptake and distribution were observed at 1 h, and day 1, 4, 7, 10, 14, and 21 after administration.

### 2.4. Wound Healing Model and Analysis

All *db/db* mice were wounded at 10 to 12 weeks old. A 1 cm diameter cutaneous full-thickness wound was excised on the mid-dorsal area. To avoid wound margins from excess macrodeformations, a 0.5 cm wide Duoderm^®^ (DuoDERM^®^, CGF^®^, ConvaTec, Squibb & Sons, L.L.C., Princeton, NJ, USA) was splinted around the wound edges [24]. Repeated administration of exosomes was performed at day 1, 4, 7, 10, 13 and 16. For each administration, ASC-Exos (200 µg) suspended in PBS (200 µL), DFb-Exos (200 µg) suspended in PBS (200 µL), or PBS (200 μL) were applied into the wounds. Wound closure was quantified by measuring contraction, re-epithelialization, and open wound as the percentage of the initial wound. The sum of contracted, re-epithelialized, and open wound areas equals 100% of the initial wound size.

### 2.5. Immunohistochemistry

Masson’s trichrome staining was performed on paraffin-embedded tissue to detect collagen fibers. To evaluate the cell proliferation and angiogenesis in wound tissues, immunohistochemistry was performed in 5 μm paraffin-embedded tissue sections. Samples were incubated with antibodies of Ki-67 (Lab Vision, Freemont, CA, USA) and PECAM-1 (Pharmingen, San Jose, CA, USA). To quantitate keratinocyte proliferation, the number of Ki-67 positive cells at the basal layer and wound was counted and expressed as a ratio of proliferating nuclei (Ki-67 positive) to total nuclei in the basal layer. To quantitate angiogenesis, the neovascular area (PECAM-1 positive cells) were measured using Image J and was expressed as a percentage of PECAM-1 positive area of the total image area.

### 2.6. Western Blotting

Wound tissues or exosomes derived from DFb and ASC of diabetic mice were homogenized and lysed in an ice-cold lysis buffer containing cocktail proteinase and phosphatase inhibitors. Equal amounts of protein extract were resolved by sodium dodecyl sulfate-polyacrylamide gel electrophoresis (SDS-PAGE) and immunoblotted. Chemiluminescent signals (LumiLight, Roche Diagnostics, Basel, Switzerland) were detected with a Lumi-Imager or X-ray films and quantified using the LumiAnalyst program (Roche Diagnostics). The supernatant of ASC-Exo treated monocytes was collected and was added into cultured dermal fibroblasts. Aliquots of conditioned media with or without the addition of SIS3 were subjected to immunoblotting with antibodies for Col-I, α-SMA, and p-Smad3. All the cells were of diabetic origin. 

### 2.7. Immunofluorescence

For immunofluorescence, monocytes and fibroblasts incubated with R18 labeled ASC-Exo were grown on Labtek II glass slides and were fixed and permeabilized. After blocking with 5% BSA/PBS, Cy5 conjugated antibody to phalloidin was added overnight at 4 °C in 5% BSA/PBS. The FITC-labeled secondary anti-goat was added and cells were incubated with 4′,6-diamidino-2-phenylindole (DAPI). Coverslips were mounted on glass slides with glycerin and analyzed using Nikon fluorescence microscopy.

### 2.8. Statistics

Statistical analyses were performed using SPSS 20.0. All statistical data are presented as mean ± SE. Comparisons between two groups were analyzed using independent unpaired two-tailed Student’s *t* tests, and comparisons between more than two groups were analyzed using one-way analysis of variance (ANOVA) with Bonferroni correction. *p* values less than 0.05 were considered statistically significant.

## 3. Results

### 3.1. ASC-Exo has a Higher Amount of Total Protein than DFb-Exo

Exosomes from the culture supernatant of diabetic ASCs and DFb were isolated via differential centrifugation (Figure 1B). We found that ASCs secreted higher amounts of exosome-associated proteins in the culture supernatant compared to DFb (Figure 1G). Confirming previously reported results, the morphology of isolated exosomes, as assessed by transmission electron microscopy, showed vesicles in typically associated doubled-layered cup-shaped morphology with the size of around 85 nm (Figure 1C). The size of ASC-Exo and DFb-Exo was directly measured by using the NanoSight system. We observed the mean size of ASC-Exo and DFb-Exo was 119.1 ± 27.2 and 110.2 ± 23.5 nm, respectively, which was consistent with the size of the exosome to be 30–150 nm in previous reports (Figure 1D). We verified that exosomes in both groups were enriched in membrane-associated tetraspanins CD9, CD63, and CD81 by Western blotting (Figure 1E). Specifically, The ASC-Exo has a greater CD63 enrichment, while the DFb-Exo has a greater CD9 and CD81 ratio (Figure 1F).

### 3.2. Spatial and Temporal Biodistribution of Topically Administered Exosomes In Vivo

To elucidate the role of the ability of ASC-Exo to target the wound site, it is of utmost importance to firstly determine the in vivo fate of exosomes. Here, we studied the biodistribution of R18-labeled exosomes in diabetic mice after topical delivery to assess if differences in cell types can influence the uptake and tissue biodistribution. The fluorescence intensity of the lesion region was significantly higher after R18-labeled exosomes administration as compared to PBS and showed a time-dependent attenuation till 21 days (Figure 2A). Notably, the fluorescence intensity increased dramatically to reach a peak value 1 h and rapidly dequenched 3 days following exosome administration in the ASC-Exo and DFb-Exo treated wounds (Figure 2B). Moreover, fluorescence quantification determined a significant accumulation of ASC-Exo and DFb-Exo in wound sites. Overall, this suggests that topically applied exosomes could efficiently target and preferentially accumulate in wound tissue and the cellular origin, ASC or DFb, has no influence on the biodistribution pattern of exosomes.

### 3.3. ASC-Exo Enhance Wound Closure by Contraction and Re-Epithelialization

Topical delivery of ASC-Exo induced faster wound closure over time compared with treatment with DFb-Exo and PBS (Figure 3B,C). This difference in the rates of wound closure in diabetic mice between those treated with ASC-Exo and those with either DFb-Exo or PBS reached statistical significance by day 10 (Figure 3C). Contraction in the ASC-Exo treated wounds was increased compared with the DFb-Exo or PBS-treated wounds on days 10, 14, and 17 (Figure 3D). Wounds treated with ASC-Exo showed augmented re-epithelialization compared with the DFb-Exo and PBS groups on day 14 and 17 (Figure 3E).

### 3.4. ASC-Exo Upregulate Cell Proliferation and Angiogenesis in Wound Sites

The wounds treated with ASC-Exo had a significantly thicker granulation tissue and collagen deposition than those in the DFb-Exo and PBS groups on day 17 (ASC-Exo vs. DFb-Exo, *p* < 0.001; ASC-Exo vs. PBS, *p* < 0.001) (Figure 4A–C). To investigate whether ASC-Exo affected blood vessel formation and active cell proliferation in wound sites, immunohistochemistry of wounds was performed and quantified. Proliferating cells in the basal layer and wound bed were stained for Ki-67, and PECAM-1 (CD31) was selected to stain the endothelial cells (Figure 4D). The Ki-67 signal within the basal layer and granulation tissue was significantly enhanced in the groups with DFb-Exo and PBS (Figure 4E,F). Most of the actively proliferating cells populating in wound bed were fibroblasts by morphology and Masson’s trichrome stain. The ASC-Exo-treated wounds showed significant increases in blood vessel density as compared to DFb-Exo and PBS groups (ASC-Exo vs. DFb-Exo, *p* < 0.001; ASC-Exo vs. PBS, *p* < 0.001) (Figure 4G).

### 3.5. ASC-Exo Upregulate Protein Expression Related to Proinflammatory Chemokines, Angiogenesis, Re-Epithelialization, and ECM Remodeling in Wounded Tissue

As shown in Figure 5, the stromal cell-derived factor (SDF)-1 were significantly upregulated in ASC-Exo-treated wounds compared with DFb-Exo and PBS-treated wounds. The protein expression of VEGF was significantly higher and showed a correlation with increased numbers of blood vessel density in ASC-Exo-treated wounds than DFb-Exo and PBS-treated wounds (Figure 5E). However, no or very weak protein expression of b-FGF was detected throughout all the three groups. ASC-Exo-treated wounds also showed a 2.5-fold and a 1.4-fold increase in expression of keratinocyte growth factor (KGF), a potent epithelial cell-specific growth factor, as compared with PBS and DFb-Exo-treated groups, respectively (Figure 5D). ASC-Exo-treated wounds resulted in a significantly increased protein expression of Col-I and α-SMA when compared with DFb-Exo and PBS-treated wounds. Smad3 functions as a transcriptional modulator to regulate the TGF-β signaling pathway in collagen deposition and tissue fibrosis. Here, the protein expression of Smad3 was significantly higher in the ASC-Exo treated wounds and showed a correlation with increased expression of Col-I and α-SMA at wound sites on day 17.

### 3.6. ASC-Exo Promotes Collagen Production through TGF-β/Smad3 Signaling Pathway

In order to test the uptake of ASC-Exo by monocytes or dermal fibroblasts, we incubated R18 labeled ASC-Exo with monocyte and dermal fibroblasts at different time points and examined the internalization by confocal microscopy analysis. ASC-Exo internalization was observed as early as 2 h post-incubation. A higher accumulation of exosomes inside monocytes or fibroblasts was found at 12 h post-incubation (Figure 6A,B). To investigate TGF-β signaling and to test the effects of ASC-Exo on different cell types in the current study, the TGF-β1 protein levels were examined on cultured monocytes and fibroblasts using ELISA. The TGF-β1 levels of ASC-Exo-treated monocytes and fibroblasts were markedly increased than the non-treated groups on day 3 post-incubation (Figure 6C). Next, we evaluated SIS3, a potent and selective inhibitor of Smad3 function, in the suppression of the TGF-β1 induced Col-I and α-SMA up-regulation in fibroblasts. We found that the addition of SIS3 diminished the protein expression of Col-I and α-SMA (Figure 6D–H).

## 4. Discussion

Despite the initial promise, it is increasingly clear that utilities of cell therapy are limited by a host of concerns, the least of which involves cross-host immunogenicity. Exosomes, known to be minimally immunogenic and cross-host capable, offer great potential in regenerative medicine as a cell-free alternative to cell therapy. In our recent study, ASC-Exo demonstrated comparable effects to their source cells in fat graft retention [25]. An ideal source cell origin for exosomes should be of high regenerative potential with an easily accessible and abundant donor source. To that goal, we chose ASCs as the cell source to examine the therapeutic potential in a diabetic wound healing model with the residential DFb as the pertinent cell control. Our study has demonstrated that topical administration can be an efficient means to deliver functional exosomes and that ASC-Exo can accelerate diabetic wound healing by stimulating cell proliferation, re-epithelialization, contraction, and angiogenesis. Furthermore, for the first time, we have elucidated that a main mechanism through which ASC-Exo improved wound regeneration would involve activation of the TGF-β/Smad3 signaling pathway, which promotes collagen I production.

In our study, differential centrifugation was chosen as the methodology to isolate exosomes. We went on to verify their identity through TEM, NTA, and expression profiles of uniquely characteristic surface tetraspanins CD9, CD63, or CD81. While different tetraspanin expressions were noted between ASC- and DFb-derived exosomes, total protein expression was distinctively higher in the ASC-Exo group, indicating the likelihood of higher activities with those exosomes. It is, therefore, particularly interesting that one such tetraspanin, CD63, which has been implicated previously to correlate directly with exosome production and quantity, was significantly elevated with the ASC-Exo group compared to those from the DFb group [10,26,27,28]. Further, our NTA study seemed to corroborate that ASC-Exo were in general larger vesicles than their DFb-derived counterpart, despite both having a relatively wide ranging and heterogeneous size distribution (Figure 1). Collectively, these results indicate that ASCs were more active in secreting a higher number of exosomes compared with DFb in the same condition.

Exosomes are highly prevalent among cells in culture and body fluid and can likely serve as important messengers of intercellular communication. The trafficking of secretory exosomes and regulation of membrane fusion are mediated by several mechanisms, including endocytosis, phagocytosis, micropinocytosis, lipid raft-mediated internalization, and direct fusion [29]. Octadecyl rhodamine B chloride (R18) is a lipophilic and fluorescent dye ideal for membrane fusion assay [30]. In this study, ASCs or DFb-derived exosomes were labeled with R18 to monitor the uptake and biodistribution upon topical administration in a diabetic wound mice model using an IVIS imaging system. As we only labeled the first application of exosomes with R18 (Figure 2), we were able to trace the progression of the fluorescent signals over time as a surrogate for exosome uptake. Although the radiant efficiency of exosome uptake in wound tissue showed an overall time-dependent attenuation till 21 days after treatment, a more linear and gradual signal decline commenced about one day after the exosome application after an initial precipitous drop that is most likely related to the washout effects of the application itself. This attenuation pattern was seen in both the ASC-Exo and DFb-Exo treated wounds, suggestive of a consistent exosome uptake time course and potentially the basis for any future topical therapeutic applications.

Wound regeneration is a dynamic process that synergistically combines regenerative cells, hospitable wound matrix/materials, and growth factors in three continuous and overlapping stages: an inflammatory phase, a proliferative phase, and a contraction and remodeling phase. In our study, ASC-Exo-treated wounds healed significantly faster than those treated with ASC-DFb or PBS controls. This healing advantage seemed to be rooted through both epithelialization and contraction, as the two components were analyzed independently. These histological observations were corroborated by an up-regulation of Ki-67 expression in both basal keratinocytes and fibroblasts, respectively. These findings at the cellular level were strongly correlated at the molecular level as well. In the epidermis, a significant increase in levels of KGF was detected by Western blot with the ASC-Exo treatment. Dermal cell proliferation, already manifested with increased granulation thickness and collagen deposition in ASC-Exo-treated wounds, were further supported by increased Collagen I and α-SMA at the protein level. Interestingly, statistically significant increases in these factors were also noted with the DFb-Exo, although much less so than with the ASC-Exo-treated wound only at the molecular level. This seemed to suggest that ASC-Exo was capable of delivering significant clinical effects above a threshold that DFb-Exo could not match. Taken together, we have provided the evidence of topical administration of diabetic ASCs-derived exosomes to enhance cutaneous wound healing by a significant acceleration in macroscopic wound closure, contraction, and re-epithelialization through their profound stimulation of resident fibroblasts/myofibroblasts as well as keratinocytes as compared to diabetic fibroblast-derived exosomes.

Angiogenesis is known to play a central role in wound regeneration and b-FGF and VEGF are the major growth factors involved. A previous study showed diabetes-related depletion of a subpopulation of ASCs characterized by its pro-vascular transcriptional profile compromised angiogenic potential with a limited regenerative capacity [31]. Seemingly, our results using diabetic ASC-derived exosomes would counter the current dogma on the impairment of diabetic ASC functionality with reduced therapeutic potential for wound neovascularization but may also shed light on a potential bypass of a major hurdle in utilizing cells from a diabetic background for regenerative purposes. The significantly higher protein expression of VEGF observed in the ASC-Exo-treated wounds correlated with wound tissue stained for PECAM-1. Interestingly, no or very weak expression of b-FGF was detected by Western blot. As compared to diabetic DFb-Exo, it might be speculated that the diabetic ASC-Exo could increase angiogenesis and vascularity in wounds via VEGF dependent pathway. Alternatively, initial ASC isolation prior to exosome purification in a conditioned medium may have modified the expression profile and thus the genesis of those exosomes from the diabetic ASCs. Future investigations are certainly needed to ascertain these important aspects of regenerative vehicles of a diabetic background.

In vitro, ASC-Exo markedly increases TGF-β1 production in cultured monocytes, which demonstrated that monocytes or macrophages undergo marked functional changes and initiate their production and activation of the profibrotic cytokine TGF-β1. This result is consistent with a significantly higher protein expression of TGF-β1and α-SMA in ASC-Exo-treated wounds. Among the complicated signaling pathways involved in collagen production and tissue fibrosis, the TGF-β/Smad3 pathway has been widely recognized. It is well accepted that TGF-β binds to the receptors on fibroblasts, thereby activating the Smad3 complex to facilitate its nuclear transcription and initiating the synthesis and secretion of collagen and α-SMA. As Smad3 is believed to be the final downstream mediator of the signaling pathway, it can directly modulate ECM-involved gene expression. In our study, the blockade of Smad3 by its inhibitor SIS3 significantly suppressed ECM production, as evidenced by decreased protein levels of Col-1 and α-SMA. Our finding implicated that the exogeneous ASC-Exo were involved in the crosstalk between the dermal fibroblasts and monocytes/macrophages of the wound-site microenvironment. By adding ASC-Exo, it stimulated resident monocytes/macrophages to secrete more TGF-β1 and activate the TGF-β/Smad3 signaling pathway, which then led to increased synthesis of collagen and other components of ECM by fibroblasts to enhance wound healing. Of note, ASC-Exo could increase the production of TGF-β1 in fibroblasts. TGF-β1 transported to fibroblasts by exosomes constitutes a rapid response and those fibroblasts were activated to initiate the production of TGF-β1 protein in an autocrine fashion, which leads to more proliferation and activation of fibroblasts. Therefore, we propose that TGF-β1 is centrally involved in ASC-Exo mediated cellular crosstalk as an important early response to initiating wound regeneration (Figure 7).

## 5. Conclusions

Our data showed that diabetic ASC-Exo had a superior cell proliferation, re-epithelialization, contraction, and angiogenesis compared to those from diabetic DFb-Exo in diabetic wound healing, of which TGF-β1 played a central role in this healing cascade. The results also inform the potential utility of the cell-free exosomal approach from diabetic donors in diabetic wound healing.

## Figures and Tables

**Figure 1 pharmaceutics-14-01206-f001:**
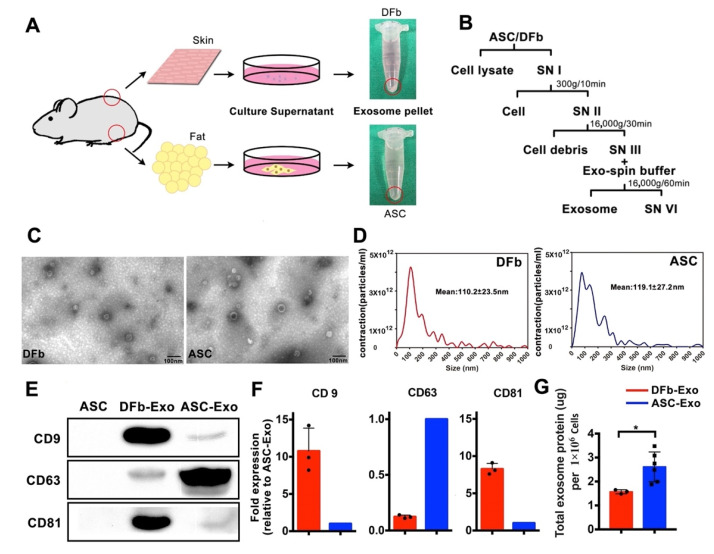
Characterization of dermal fibroblast (DFb) and adipose stem cell (ASC) derived exosomes. (**A**) Schematic illustrating the workflow used to generate exosomes. The arrows indicate harvested dermal fibroblast or adipose stem cell-derived exosome pellets; (**B**) differential centrifugation procedure for the isolation of exosomes from DFb and ASC culture supernatants (SN); (**C**) transmission electron microscopic morphological analysis of ASC-Exo and DFb-Exo depicts multiple cup-shaped and shrunken vesicles in the conditioned medium (scale bar: 100 nm); (**D**) size distribution of ASC-Exo and DFb-Exo based on NanoSight nanoparticle tracking analysis (NTA). Statistics of the size of ASC-Exo and DFb-Exo based on NTA are shown as mean ± SD (representative result of four independent analyses); (**E**) Western blot analysis of general exosome markers, including CD9, CD63, and CD81 for ASC-Exo and DFb-Exo; (**F**) semi-quantification analysis of CD63, CD9, and CD81 expression from ASC-Exo and DFb-Exo; (**G**) total protein contained within exosomes isolated from the culture supernatant of 1 × 106 dermal fibroblasts (*n* = 3) and adipose stem cells (*n* = 6). * *p* < 0.05. Error bars are mean ± SE. Data were analyzed using independent unpaired two-tailed Student’s *t*-tests.

**Figure 2 pharmaceutics-14-01206-f002:**
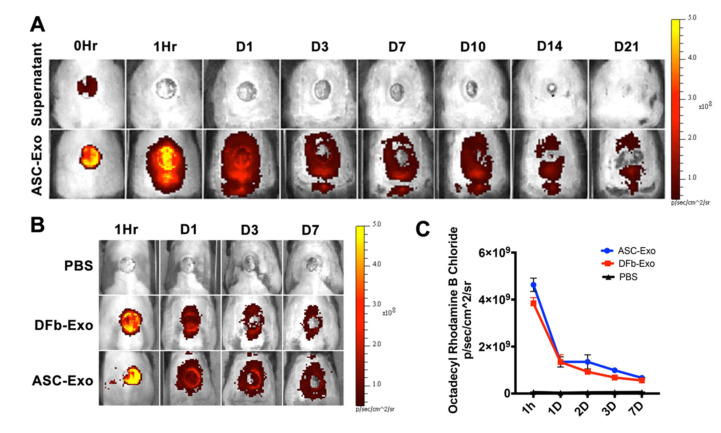
In vivo tracking of exosomes attachment onto wound tissue. (**A**) The bioluminescence distribution of octadecyl rhodamine B chloride (R18)-labeled ASC-Exo during uptake demonstrated strong fluorescent signal within 1 h and showed a time-dependent attenuation till 21 days after treatment. (**B**) The representative images of uptake of ASC-Exo, DFb-Exo, and PBS at wound sites within 7 days post wounding. (**C**) Comparison of fluorescence intensity throughout all groups.

**Figure 3 pharmaceutics-14-01206-f003:**
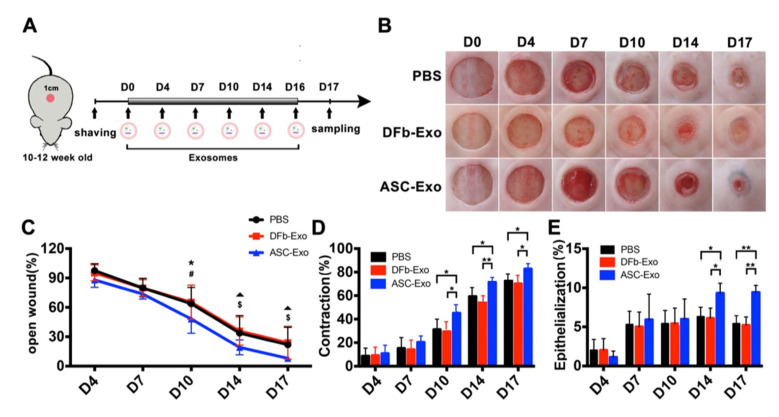
Wound healing kinetics in PBS, dermal fibroblast (DFb) derived exosomes, and adipose stem cell (ASC) derived exosomes treated wounds. (**A**) Scheme illustrating the experimental procedure and treatment in db/db mice; (**B**) representative macroscopic images and quantification of raw surface, re-epithelialization, and contraction. (*n* = 8/group); (**C**) time-course changes of raw surface revealed significant differences were found on day 10, 14 and 17 (* compare with DFb or PBS, *p* < 0.05; # compare with DFb or PBS, *p* < 0.01); (**D**) enhanced wound contraction was found in ASC-Exo treated wounds on day 10, 14 and 17 (* *p* < 0.05; ** *p* < 0.01); (**E**) re-epithelialization was enhanced in ASC-Exo treated wounds when compared with DFb-Exo and PBS group on day 14 and 17 (** *p* < 0.01). Error bars are mean ± SE. Data were analyzed using independent unpaired two-tailed Student’s *t*-tests.

**Figure 4 pharmaceutics-14-01206-f004:**
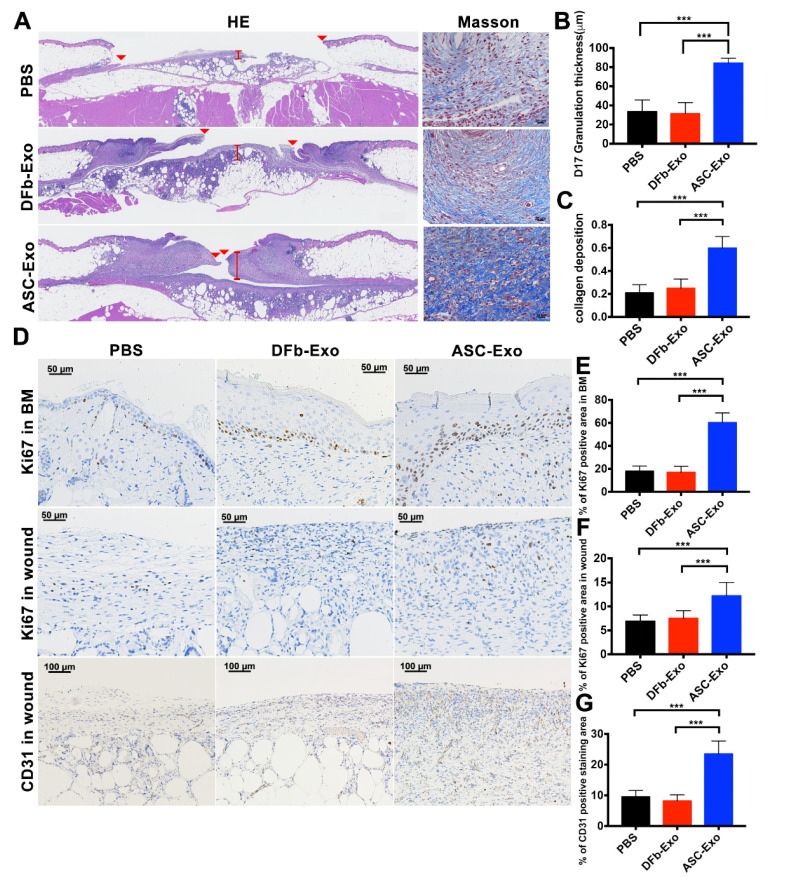
ASC-Exo increases basal keratinocyte and dermal cell proliferation and angiogenesis in wound healing. (**A**) Representative histology on 17 days after wounding for each group. Left panel, hematoxylin and eosin (H&E) stain. The wounds treated with ASC-Exo showed a significantly thicker granulation tissue in comparison with the DFb-Exo and PBS group. Red arrows indicate tips of the healing epithelium tongue. Right panel, Masson’s trichrome stain (scale bar: 1 mm). (**B**) Wound healing tissues were stained for basal keratinocyte and dermal cell proliferation (Ki-67) (upper and central panel, magnification: 200×) and vascularity (PECAM-1) (lower panel, magnification: 100×). (**C**,**D**) The wounds treated with ASC-Exo showed a significantly thicker granulation tissue and collage deposition in comparison with the DFb-Exo and PBS group. (**E**,**F**) The ASC-Exo-treated wounds showed significant increases in basal keratinocytes and dermal cell proliferation when compared with both the DFb-Exo and PBS groups on day 17 (*** *p* < 0.001). (**G**) The ASC-Exo-treated wounds showed significant increases in calculated blood vessel density when compared with both the DFb-Exo and PBS groups on day 17 (*** *p* < 0.001). Results represent mean ± SE. Data were analyzed using independent unpaired two-tailed Student’s *t*-tests (**C**–**G**).

**Figure 5 pharmaceutics-14-01206-f005:**
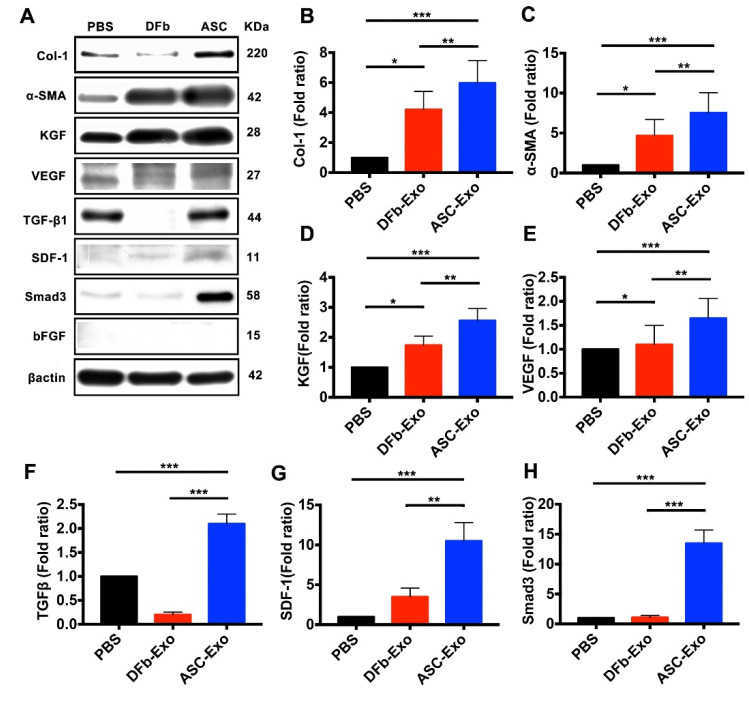
(**A**–**H**) In vivo Western blot analysis of the protein expression comparing wounds on day 17 from ASC-Exo, DFb-Exo, and PBS-treated mice. Data were standardized by the expression level of β-actin in each sample and presented as the relative expression of those from the PBS group (* *p* < 0.05; ** *p* < 0.01; *** *p* < 0.001). One representative of five independent experiments is shown. Results represent mean ± SE. Data were analyzed using independent unpaired two-tailed Student’s *t*-tests.

**Figure 6 pharmaceutics-14-01206-f006:**
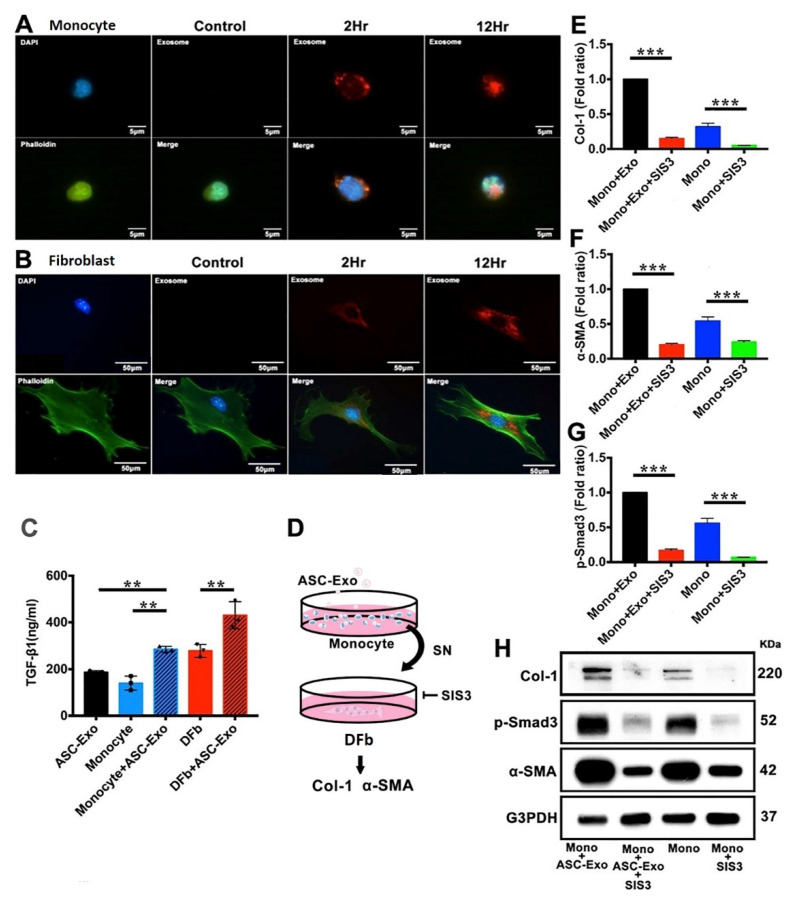
Cell internalization of ASC-Exo in cultured monocytes and dermal fibroblasts (DFb) of diabetic mice and role of ASC-Exo in Smad3-dependent TGF-β stimulation of ECM production. (**A**) Cellular uptake of ASC-Exo in cultured monocytes by confocal microscopy demonstrated rapid uptake of ASC-Exo. Merged images of DAPI-labelled nucleus (blue), R18-labelled ASC-Exo (red), and phalloidin-labelled cytoskeleton (green) revealed localization of ASC-Exo around the cells at 2 h and into the cells at 12 h. (**B**) ASC-Exo uptake by cultured dermal fibroblasts. (**C**) ASC-Exo treated monocytes and dermal fibroblasts increase the production TGF-β1. TGF-β1 levels were measured using enzyme-linked immunosorbent assay (** *p* < 0.01). Data were expressed in means ± SE, *n* = 3/group, Student’s *t*-tests. (**D**) Experimental design of SIS3, a specific inhibitor of Smad3, on TGF-β1 dependent Smad3 phosphorylation and extracellular matrix production in cultured DFb. The DFb cultures were serum-starved for 24 h and treated with 3 µM SIS3 for 1 h, and then supernatant of ASC-Exo-treated monocyte culture was added. After 48 h, the conditioned media were subjected to immunoblotting with antibodies for Col-I, α-SMA, p-Smad3, and β-actin, respectively. One representative of four independent experiments is shown. Error bars are mean ± SE. Data were analyzed using independent unpaired two-tailed Student’s *t*-tests or 1-way ANOVA with Bonferroni correction. (**E**–**H**) SIS3 inhibited the protein expression of Col-I, α-SMA, and p-Smad3 induced by supernatant of ASC-Exo-treated monocyte culture in dermal fibroblasts. Data are expressed as the mean ± SE of four independent experiments. *** *p* < 0.001 compared with the value in the supernatant of ASC-Exo-treated monocyte culture treated DFb without SIS3.

**Figure 7 pharmaceutics-14-01206-f007:**
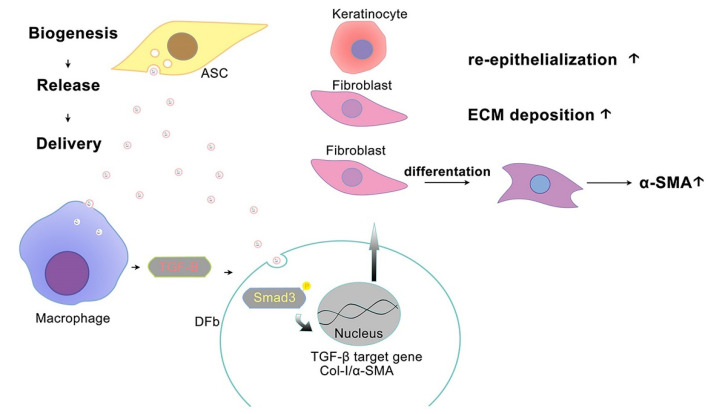
A proposed mechanism depicting the pathway by which diabetic ASC-Exo enhances diabetic wound healing. By adding ASC-Exo, resident monocytes/macrophages were stimulated to secrete more TGF-β1 and activate the TGF-β/Smad3 signaling pathway. Of note, ASC-Exo could increase the production of TGF-β1 of fibroblasts. Collectively, these findings indicate that TGF-β1 is centrally involved in ASC-Exo mediated cellular crosstalk as an important early response to initiate wound regeneration.

## Data Availability

The data that support the findings of this study are available from the corresponding author upon reasonable request.

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
