# Peer review of "Therapeutic Potential of Exosomes Derived from Diabetic Adipose Stem Cells in Cutaneous Wound Healing of db/db Mice"

_pharmaceutics, 2022, doi:10.3390/pharmaceutics14061206_

Round 1

Reviewer 1 Report

Hsu et al demonstrated that exosomes isolated from adipose stem cells (ASC) and dermal fibroblasts accelerated the healing of skin wounds in db/db mice. This effect was accompanied by the stimulation of cell proliferation, angiogenesis and production of SDF and KGF. Un vitro, exosomes enhanced the production of TGFbeta in monocytes, and medium conditioned by monocytes stimulated in TGF-dependent manner the production of collagen 1 and smooth muscle actin in the culture of dermal fibroblasts. The authors suggest that exosomes could enhance wound healing through a TGFbeta dependent mechanism.

Considering the abundance of published studies about the wound healing effects of exosomes, the study does present a sufficient element of novelty. Moreover, it is rather superficial. The authors did not make an attempt to understand which molecular components of exosomes are responsible for the stimulation of TGFbeta expression in monocytes. They could use confocal immunofluorescence microscopy to demonstrate the exosome-dependent stimulation of TGFbeta expression in monocytes/macrophages in the wound. They also could assess the dependence of exosome-stimulated wound healing on TGFbeta signaling.

Author Response

Q. Considering the abundance of published studies about the wound healing effects of exosomes, the study does present a sufficient element of novelty. Moreover, it is rather superficial. The authors did not make an attempt to understand which molecular components of exosomes are responsible for the stimulation of TGF beta expression in monocytes. They could use confocal immunofluorescence microscopy to demonstrate the exosome-dependent stimulation of TGF beta expression in monocytes/macrophages in the wound. They also could assess the dependence of exosome-stimulated wound healing on TGF beta signaling.

Response : Thank you for your comments. We do appreciate your comments on this matter. However, could we attempt to answer your query with the results shown in Fig. 5 and 6C where TGF beta expression of the western blot result in ASC-Exo treated wound was statistically higher and also of a higher level of TGF beta in the cultured monocytes treated with ASC-Exo in Fig. 6C. ASC-Exo markedly increases TGF-β1 production in cultured monocytes, which demonstrated that monocytes/macrophages undergo marked functional stimulation and initiate their production and activation of the profibrotic cytokine TGF-β1. This result is consistent with a significantly higher protein expression of TGF-β1 of ASC-Exo treated wound. These combined results could provide evidence of exosome dependent stimulation of TGF beta expression in monocytes/macrophages in the wound.  

Reviewer 2 Report

There is one reference worthy to consider to add: Kalluri R, LeBleu V. The biology, function, and biomedical applications of exosomes. Science, 2020;367 (6478).

Excellent paper.

Author Response

Q. There is one reference worthy to consider to add: Kalluri R, LeBleu V. The biology, function, and biomedical applications of exosomes. Science, 2020;367 (6478).

Response: We thank the reviewer for their appreciation of our work and suggestions proposed . We will add this to our reference section as reference 12.

Reviewer 3 Report

Interesting article in the area of study the in vivo mechanistic basis by which diabetic ASC-Exo enhance cutaneous wound healing in a diabetic mouse model. The search for effective therapy for chronic cutaneous wound is very relevant. The article is recommended for publication.

Author Response

We would like to thank the reviewer for their time to go through our article and their appreciation of our work as well as recommendation for publication.

Reviewer 4 Report

In this study, the authors examined the potential of diabetic ASC-derived exosomes as cell-free therapeutics for diabetic wound healing. They demonstrated that diabetic ASCs-derived exosomes can enhance healing of wounds in diabetic mice. The authors identified that the ASC-exosomes mediate the healing via increases in pro-angiogenic growth factors and wound contraction proteins, and induction of TGF-1β/Smad3 signalling pathway. Overall, the article reads well. The findings of this study may help innovate superior strategies for diabetic wound healing. However, certain useful information appears to be missing that can further improve the article. The authors are hereby requested to address the following issues:

Major comments:

  1. It is essential that we understand the therapeutic potential of diabetic ASC-exosomes. However, it isn’t described sufficiently why the authors chose to use diabetic ASC-exosomes. Please elaborate on the rationale for the use of diabetic ASC-exosomes.
  2. In the Introduction, it would be beneficial to discuss briefly any findings of the therapeutic efficacies between allogenic and autologous ASC-derived exosomes on wound healing. Also, to discuss what impact does stem cell-derived exosomes have on wound inflammation.

  1. In the M&M, what tests were performed to confirm diabetes in the mice? Please provide basic animal data such as the body weight and blood glucose level for both the 10 weeks and 12 weeks old mice.

  1. Cells isolated from genetically diabetic hosts are generally assumed to carry the diabetic traits and used as such for studies. However, prior to use, I feel it is important to confirm that the cells express diabetic traits by comparing the expression of diabetes-associated markers such as advanced glycation end products (a critical by-product of hyperglycemia) against non-diabetic cells. Alternatively, proteome profiling may be used. This may help identify the defect in the diabetic cells used.

  1. In the Discussion section, 5th paragraph, it can be agreeable that the diabetic ASC-derived exosomes is a potential alternative to the ASCs itself. However, without a direct comparison between the diabetic ASCs, the claim that the diabetic ASC-derived exosomes is functionally superior is a little too strong to be made in the current paper.

  1. Inflammation is one of the major issues in diabetic wounds. Inclusion of the histological assessment of pro- or anti-inflammatory immune cell populations in the wound bed will significantly add to the value of the paper.

  1. The experimental set up to elucidate the TGF-β1/Smad3 pathway was not described in the M&M. Are the monocytes and dermal fibroblasts of diabetic origin?

  1. TGF-β1 was induced in the wound by the exosomes not delivered. Please revise the title.

Minor comments:

  1. In the M&M, “Isolation and identification of exosomes” section, please provide reference to the protocol the authors used to isolate ASCs and DFbs from the mice. Also, please check the centrifugation speed 3000g to spin down cells. I believe it should be 300g.

  1. Figure 3, the time points for wound assessment shown in images A and B does not correlate with the time points shown in the graphs C, D and E. Also, the statistical data presented in graphs D and E is not convincing. Please revise.

  1. Figure 4, Image B, please provide the “unit” of granulation thickness measurement. Image C, typo error in the Y-axis label.

  1. Figure 4, Images E and F (Ki-67), I believe the data is presented as percentage not ratio. Please revise.

Author Response

In this study, the authors examined the potential of diabetic ASC-derived exosomes as cell-free therapeutics for diabetic wound healing. They demonstrated that diabetic ASCs-derived exosomes can enhance healing of wounds in diabetic mice. The authors identified that the ASC-exosomes mediate the healing via increases in pro-angiogenic growth factors and wound contraction proteins, and induction of TGF-1β/Smad3 signaling pathway. Overall, the article reads well. The findings of this study may help innovate superior strategies for diabetic wound healing. However, certain useful information appears to be missing that can further improve the article. The authors are hereby requested to address the following issues:

Major comments:

Q1. It is essential that we understand the therapeutic potential of diabetic ASC-exosomes. However, it isn’t described sufficiently why the authors chose to use diabetic ASC-exosomes. Please elaborate on the rationale for the use of diabetic ASC-exosomes.

Response : Thank you for your comments. The choice of using ASC and ASC-Exo from diabetic mice is to reflect the possible clinical nature of this implication where autologous cell therapy could potentially be used in this subgroup of diabetic patients. Furthermore, others in the literature have also utilized ASCs from diabetic population and provided its therapeutic potential in insulin resistance or wound healing. (Wang M, Song L, Strange C, Dong X, Wang H. Therapeutic Effects of Adipose Stem Cells from Diabetic Mice for the Treatment of Type 2 Diabetes. Mol Ther. 2018 Aug 1;26(8):1921-1930 and Sun Y, Song L, Zhang Y, Wang H, Dong X. Adipose stem cells from type 2 diabetic mice exhibit therapeutic potential in wound healing. Stem Cell Res Ther. 2020 Jul 17;11(1):298.) In the study, we assessed and compared the effectiveness of using ASC-Exo or DFb-Exo from the diabetic mice. This could facilitate the understanding of the mechanism of action and potential clinical use of autologous ASC-Exo in diabetic population.

This rationale has been added into the introduction section and highlighted in red color in page 5, line 15 to page 6 line 6. 

Q2. In the Introduction, it would be beneficial to discuss briefly any findings of the therapeutic efficacies between allogenic and autologous ASC-derived exosomes on wound healing. Also, to discuss what impact does stem cell-derived exosomes have on wound inflammation.

Response : Compared to ASCs, ASC-Exo might avoid immunogenicity associated problems and studies have shown that ASC-Exo could enhance cutaneous wound healing through a different signal pathway and mechanism. Till now, ASC-Exo lacked enough evidence to compare its effectiveness between autologous or allogenic origin. However, Lu et al. demonstrated that allogeneic iPSCs derived exosomes provided less effectiveness in wound healing as compared to the autologous origin. We have added the above mentioned into the introduction section in page 5, line 10-14.

The inflammatory response plays an important role in wound healing and the macrophage is a prominent inflammatory cell. Emerging evidence suggests that ASC-Exo could regulate M2 macrophage polarization to mitigate the inflammation. 

Q3. In the M&M, what tests were performed to confirm diabetes in the mice? Please provide basic animal data such as the body weight and blood glucose level for both the 10 weeks and 12 weeks old mice.

Response: The body weight of db/db mice ranged from 40 to 43 g and the blood sugar ranged from 515 to 530 mg/dl at the age of 10 to 12 weeks old.

Image in the uploaded document.

Q4. Cells isolated from genetically diabetic hosts are generally assumed to carry the diabetic traits and used as such for studies. However, prior to use, I feel it is important to confirm that the cells express diabetic traits by comparing the expression of diabetes-associated markers such as advanced glycation end products (a critical by-product of hyperglycemia) against non-diabetic cells. Alternatively, proteome profiling may be used. This may help identify the defect in the diabetic cells used.

Response : Thank you for your query. This mice model has already been validated in our prior published paper and we can supply those as supplementary evidence.

  1.  Chen B, Kao HK, Dong Z, Jiang Z, Guo L. Complementary Effects of Negative-Pressure Wound Therapy and Pulse Radiofrequency Energy on Cutaneous Wound Healing in Diabetic Mice. Plast Reconstr Surg. 2017; 139: 105-117.
  2. Kao HK, Li Q, Flynn B, Qiao X, Ruberti JW, Murphy JF, Guo L*. Collagen synthesis modulated in wounds treated by pulsed radiofrequency energy. Plast Reconstr Surg. 2013; 131(4): 490e-498e. 
  3. Kao HK, Chen B, Murphy GF, Li, Q,Orgill DS, Guo LF*. Peripheral blood fibrocytes: enhancement of wound healing by cell proliferation, re-epithelialization, contraction, and angiogenesis. Ann Surg. 2011 Dec; 254(6): 1066-74. 
  4. Peng C, Chen B, Kao HK, Murphy G, Orgill DP, Guo L*. Lack of FGF-7 further delays cutaneous wound healing in diabetic mice. Plast Reconstr Surg. 2011 Dec; 128(6): 674e-683e. 
  5. Li Q, Kao HK, Matros E, Peng C, Murphy GF, Guo, LF*. Pulsed Radio Frequency Energy (PRFE) Accelerates Wound Healing In Diabetic Mice. Plast Reconstr Surg. 2011 Jun; 127(6):2255-62. 

Q5. In the Discussion section, 5th paragraph, it can be agreeable that the diabetic ASC-derived exosomes is a potential alternative to the ASCs itself. However, without a direct comparison between the diabetic ASCs, the claim that the diabetic ASC-derived exosomes is functionally superior is a little too strong to be made in the current paper.

Response: Thanks for the comments. It has been revised and please see the highlighted red color in page 20. 

 Q6. Inflammation is one of the major issues in diabetic wounds. Inclusion of the histological assessment of pro- or anti-inflammatory immune cell populations in the wound bed will significantly add to the value of the paper.

Response: Thank you for your query. In the experimental design, repeated administration of exosomes was performed. We obtained histological samples at day 17 when one of the groups reached 90% of wound healing. Hence, we did not expect a raised inflammatory response at day 17 as that would be at the regenerative phase of wound healing. If a difference in inflammation were to be seen it would be hypothesized that a difference is best observed during the inflammation period from roughly day 3-7. 

Q7. The experimental set up to elucidate the TGF-β1/Smad3 pathway was not described in the M&M. Are the monocytes and dermal fibroblasts of diabetic origin?

 Response: (1) Thanks for the comment. We have added the experimental set up into the method section and highlighted in red color in page 10, line 10-14. (2) The monocytes and fibroblasts are diabetic origin.

Q8. TGF-β1 was induced in the wound by the exosomes not delivered. Please revise the title.

Response: We thank the reviewer’s comment. In order to avoid any misunderstanding , we have revised the title as “Therapeutic Potential of Exosomes Derived from Diabetic Adipose Stem Cells in Cutaneous Wound Healing of db/db Mice”. 

Minor comments:

Q1. In the M&M, “Isolation and identification of exosomes” section, please provide reference to the protocol the authors used to isolate ASCs and DFbs from the mice. Also, please check the centrifugation speed 3000g to spin down cells. I believe it should be 300g.

Response: Thanks. We have revised it. 

Q2. Figure 3, the time points for wound assessment shown in images A and B does not correlate with the time points shown in the graphs C, D and E. Also, the statistical data presented in graphs D and E is not convincing. Please revise.

Response: Thanks. It has been revised.

Q3. Figure 4, Image B, please provide the “unit” of granulation thickness measurement. Image C, typo error in the Y-axis label.

Response: Thanks. It has been revised.

Q4. Figure 4, Images E and F (Ki-67), I believe the data is presented as percentage not ratio. Please revise.

Response: Thanks. It has been revised.

Round 2

Reviewer 1 Report

Unfortunately, the authors did not provide satisfactory responses to the critiques. As originally stated, the paper in its present form fails to elucidate the mechanisms underlying the effects of exosomes.

Author Response

Thank you for your comment. In the study, our aim was to test the therapeutic effect of diabetic adipose cell derived exosomes in diabetic wound healing and the important role that TGF beta plays in it.In Fig. 6, TGF-beta was expressed in ASC-Exo and was enhanced in ASC-Exo treated monocytes and fibroblasts. The ASC-Exo derived from its diabetic donor was enriched with TGF-beta. Blocking exosome associated TGF beta stimulation with SIS3 (a specific inhibitor of TGF beta dependent Smad3 pathway) was sufficient to reduce the stimulating role of ASC-Exo in monocytes, further supporting the finding.

Exosomes are RNA, protein, lipid, and nucleic acid containing small vesicles. Therefore, RNA sequencing or proteomic analysis would be a useful approach to discover the potential molecules. There may be a variety of different molecules involved in it . Our study has shown that diabetic ASC-Exo which was enriched with TGF-Beta , which is one of the proteins in the Exosome ,enhances wound healing in the ASC-Exo treated wounds.

Reviewer 4 Report

The authors have addressed most of the queries. However, I still find some that’s not yet addressed or perhaps missed out by the authors.

Minor comments:

  1. In response to my previous query (Q5), the authors commented that the discussion section (5th para) has been revised. However, I could not find any changes as stated by the reviewer.

  1. In response to my previous query (Q6) about the inclusion of immunological histology assessment, the authors commented about “repeated administration of exosomes”. To my knowledge, this dosage was not defined in the methods. From the M&M, it gave me the impression that the application of exosomes was one-time. Please clarify.

  1. Q7 comments, a separate section in the M&M should be written for monocytes stimulation by exosomes and the subsequent studies on fibroblasts. Please include in the M&M that the cells were of diabetic origin. Non-diabetic cells may not show the same effect as these diabetic cells upon ASC-exosome treatment. It may mislead other researchers adopting the experiment.

Major comments:

  1. Referring back to Q2 in the minor comments section, I requested the authors to revise the statistical data as it was not convincing. My point being the statistical significance is quite high for the data with huge standard error. This was when the authors were showing a statistical significance of ** or p<0.01 between the control/DFb-Exo and ASC-Exo groups at days 14 and 17 in the previous draft (Figs. D & E). In the current revision, the authors revised it by changing ** to *** which I believe is more inappropriate, given the huge standard error in the control group. Therefore, I would like to request the authors to revise the data between all the groups reasonably.

Author Response

We thank the reviewer for their time and constructive suggestions . We have revised the manuscript accordingly. Please find attached the file with replies .
